# Prophylactic Treatment with Baloxavir Protects Mice from Lethal Infection with Influenza A and B Viruses

**DOI:** 10.3390/v15112264

**Published:** 2023-11-16

**Authors:** Keita Fukao, Takeshi Noshi, Shinya Shano, Kaoru Baba, Kenji Sato, Masashi Sakuramoto, Naohisa Kitade, Hideki Tanioka, Shinji Kusakabe, Takao Shishido

**Affiliations:** 1Shionogi & Co., Ltd., Osaka 561-0825, Japanshinji.kusakabe@shionogi.co.jp (S.K.); 2Shionogi TechnoAdvance Research, Co., Ltd., Osaka 561-0825, Japan

**Keywords:** influenza virus, prophylaxis, cap-dependent endonuclease inhibitor, baloxavir marboxil, baloxavir acid, oseltamivir phosphate

## Abstract

Influenza remains a worldwide health concern. Antiviral drugs are considered as one of the useful options for its prevention as a complementary measure to vaccination. Baloxavir acid selectively inhibits the cap-dependent endonuclease of influenza viruses and exhibits marked viral titre reduction in patients. Here, we describe the prophylactic potency of baloxavir acid against lethal infection with influenza A and B viruses in mice. BALB/c mice were subcutaneously administered once with baloxavir acid suspension, or orally administered once daily for 10 days with oseltamivir phosphate solution at human relevant doses. Next, the mice were intranasally inoculated with A/PR/8/34 (H1N1) or B/Hong Kong/5/72 strain at 24 to 96 h after the initial dosing. Prophylactic treatment with the antiviral drugs significantly reduced the lung viral titres and prolonged survival time. In particular, baloxavir acid showed a greater suppressive effect on lung viral titres compared to oseltamivir phosphate. In this model, baloxavir acid maintained significant prophylactic effects against influenza A and B virus infections when the plasma concentration at the time of infection was at least 0.88 and 3.58 ng/mL, respectively. The significant prophylactic efficacy observed in our mouse model suggests the potential utility of baloxavir marboxil for prophylaxis against influenza in humans.

## 1. Introduction

Influenza virus infection causes annual epidemics and is associated with 3 to 5 million cases of severe illness and 290,000 to 650,000 deaths globally [1,2]. In order to reduce the risk of severe symptoms, hospitalization and death associated with influenza, vaccination is one of the most powerful measures for prophylaxis against influenza, especially for groups at high risk for complications. However, vaccination is sometimes not sufficiently effective and is often underutilized [3,4,5]. For those who have not been vaccinated or are at high risk due to close contact with an infected individual, chemoprophylaxis with antiviral drugs can be considered. Antiviral prophylaxis is also recommended for those coming into close contact with infected individuals, such as medical workers, in order to control outbreak in households, schools, nursing homes or communities. Therefore, antiviral drugs are useful for protection from influenza virus infection as a complementary measure to vaccination.

Some neuraminidase inhibitors (NAI: zanamivir, oseltamivir, peramivir and laninamivir) have been licensed and used for treatment of influenza in various countries, and some of them are currently available for prophylactic use. Oseltamivir is taken orally at a dose of 75 mg once daily for 10 days for prophylactic use. Zanamivir is administered by two inhalations once daily for 10 days and laninamivir is given by one inhalation once daily for up to 2 days [6,7,8].

Baloxavir marboxil is the only cap-dependent endonuclease (CEN) inhibitor approved for the treatment and prevention of influenza [9,10,11]. Baloxavir acid, the active form of baloxavir marboxil, inhibits CEN activity in the PA subunit of influenza RNA polymerase that mediates the critical “cap-snatching” step of viral RNA transcription, thereby showing inhibitory potency against the replication of influenza A and B viruses [12,13,14,15,16]. In addition, it has been reported that therapeutic treatment with baloxavir marboxil exerts a significant antiviral effect, reduces an excessive inflammatory response and alveolar damage and improves survival rate in mice infected with influenza viruses [17]. Baloxavir marboxil has been licensed in Japan, the US and other countries for the treatment of acute, uncomplicated influenza A and B virus infections [18]. Although there were differences in the PK profiles due to race and body weight in clinical trials, baloxavir marboxil has been approved at a dose of 40 mg for patients weighing < 80 kg or a dose of 80 mg for those weighing ≥ 80 kg based on clinical trial results. Baloxavir marboxil was also approved for post-exposure prophylaxis against influenza A and B virus infections in Japan, the US and the EU. The results of a randomized, placebo-controlled trial for prophylaxis (BLOCKSTONE) demonstrated that the risk of laboratory-confirmed influenza was significantly lower among the household contacts in the single-dose baloxavir group than those in the placebo group [19]. As the BLOCKSTONE trial was conducted during a season of influenza A virus epidemics, the enrolled patients were mainly infected with influenza A virus. Here, we evaluated the prophylactic effect of baloxavir acid against lethal infection with influenza A and B viruses in mice.

## 2. Materials and Methods

### 2.1. Ethics

The study complied with the Standards for the Reliability of Application Data (Article 43, Enforcement Regulations, Law for the Assurance of Quality, Efficacy, and Safety of Pharmaceuticals and Medical Devices). The animal protocols were approved by the Shionogi Pharmaceutical Research Centre Institute Director (Shionogi & Co., Ltd., Toyonaka, Japan) based on the report of the Institutional Animal Care and Use Committee (Approval No. S18004D, S18009D, S19051D, S19054D and S20040D). The institutional animal facilities have been accredited by the Association for Assessment and Accreditation of Laboratory Animal Care (AAALAC) International.

### 2.2. Compounds

Baloxavir acid was synthesized by Shionogi & Co., Ltd., and oseltamivir phosphate was purchased from AK Scientific Inc. (Union City, CA, USA). Suspensions of baloxavir acid and solutions of oseltamivir phosphate were prepared with 0.5% methylcellulose 400 solution (MC, FUJIFILM Wako Pure Chemical Corporation, Osaka, Japan).

### 2.3. Viruses and Cells

A/PR/8/34 (H1N1) and B/Hong Kong/5/72 were obtained from the American Type Culture Collection (ATCC). Canine kidney MDCK cells were obtained from the European Collection of Authenticated Cell Cultures (ECACC) and maintained in minimal essential medium (MEM) (Thermo Fisher Scientific, Inc., Waltham, MA, USA) supplemented with 10% fetal bovine serum (FBS) (Sigma-Aldrich Co., Ltd., St. Louis, MO, USA) and 100 µg/mL kanamycin (Thermo Fisher Scientific, Inc.).

### 2.4. Animals

Specific-pathogen-free 6-week-old female BALB/c mice (Charles River Laboratories Japan, Inc., Yokohama, Japan) were used in this study. All mice were maintained in a temperature and relative humidity-controlled environment (20–26 °C and 30–70%, respectively), with a 12 h light and 12 h dark cycle. Standard chow diet (CE-2, CLEA Japan) and water were available ad libitum. The group assignment for treatment groups was carried out by taking body weight into account. Mice with any abnormalities in initial body weight or general condition were not included.

### 2.5. Pharmacokinetic Analysis

In this study, the pharmacokinetics experiment was conducted as an independent test from the drug efficacy experiment because repeated blood sampling affects the general condition of the animal, and as a result, appropriate drug efficacy evaluation cannot be performed. As previously reported, the half-life of baloxavir acid in serum after oral administration of baloxavir marboxil (prodrug) in rodents is considerably shorter than in humans [20,21]. Therefore, in order to mimic the time-course of the serum concentration of baloxavir acid in humans, we treated the mice with subcutaneous administration of baloxavir acid. Female BALB/c mice were subcutaneously administered baloxavir acid suspension at a dose of 1.6, 3.2 or 6.4 mg/kg. Blood samples were collected at 48, 72, 96, 120 and 192 h after single administration of baloxavir acid and then centrifuged to obtain plasma. The concentrations of baloxavir acid in plasma were determined by liquid chromatography-tandem mass spectrometry (LC-MS/MS). The lower limit of quantification of baloxavir acid in mouse plasma was 0.1 ng/mL. The values below the lower limit of quantification (BLQ) were treated as zero when the mean and standard deviation (SD) were calculated. The elimination half-life was calculated with WinNonlin software based on a non-compartment model with uniform weighting. The plasma concentrations for 10 mg/kg administration were from a previous study [20].

### 2.6. Prophylactic Treatment with Antivirals In Vivo

Female BALB/c mice (*n* = 10 per group) were subcutaneously administered once with 10 mg/kg of baloxavir acid suspension or orally administered once daily for 10 days with 5 mg/kg of oseltamivir phosphate solution. The mice were then intranasally inoculated with 4.0 × 10^2^ TCID_50_ (4 LD_50_) of A/PR/8/34 (H1N1) or 1.8 × 10^3^ TCID_50_ (2 LD_50_) of B/Hong Kong/5/72 strain at 24 h after the initial dosing under anaesthesia by the intramuscular administration of an anaesthetic solution containing medetomidine hydrochloride, midazolam and butorphanol tartrate. In order to set an appropriate inoculation viral titre as a lethal infection model, preliminary studies were conducted in advance and the inoculation viral titre is determined based on the lethal dose (LD_50_). The survival and body weight were examined daily for 21 days after the virus infection. In addition, five mice of each treatment group were euthanized, and the lung tissues were removed on days 1, 2, 4, 8 and 10 after infection. Viral titres in supernatants of lung homogenates were determined by standard TCID_50_ assay in MDCK cells [22]. The presence of cytopathic effects was determined microscopically, and the virus titres were calculated as log_10_ median tissue culture infectious doses (TCID_50_)/mL. When no cytopathic effect was observed in the lowest dilution (10-fold), the titre of undetected virus was defined as 1.5 log_10_ TCID_50_/mL. Mice were euthanized and were regarded as dead if their body weights became lower than 70% of the initial body weights according to humane endpoints.

### 2.7. Determination of the Effective Plasma Concentration of Baloxavir Acid in the Prophylactic Treatment Mouse Model

Female BALB/c mice (*n* = 10 per group) were subcutaneously administered once with 1.6, 3.2 or 6.4 mg/kg of baloxavir acid suspension. The mice were then intranasally inoculated with 4.0 × 10^2^ TCID_50_ of A/PR/8/34 (H1N1) or 1.8 × 10^3^ TCID_50_ of B/Hong Kong/5/72 strain at 48, 72 or 96 h after the dosing under anaesthesia. The survival and body weight were monitored daily for 28 days after the virus infection. Viral titres in supernatants of lung homogenates were determined (*n* = 10 per group) on days 1, 2, 4, 6, 8 and 10 after infection by standard TCID_50_ assay as described above. Mice were euthanized and were regarded as dead if their body weights became lower than 70% of the initial body weights according to humane endpoints.

### 2.8. Statistical Analysis

Differences in survival time after the virus infection were analysed by log-rank test. For the pairwise comparison of the virus titres in lungs between two groups each at each time point, the one-way analysis of variance (one-way ANOVA) model was applied by time point. In the statistical analysis for virus titres, the observations at 8 and 10 days after infection were excluded from the analysis set of the ANOVA model, because the virus titres of 4/5 mice on days 8 and 10 after infection with A/PR/8/34, and those of 4/5 mice at 8 days and all mice at 10 days after infection with B/Hong Kong/5/72, respectively, were not available for the vehicle group due to death. Multiplicity was adjusted by the Tukey method or fixed-sequence procedure. Statistical analysis was performed using the statistical analysis software SAS for Windows (SAS Institute, Cary, NC, USA).

### 2.9. Hemagglutination Inhibition Testing

Blood was collected from mice infected with A/PR/8/34 (H1N1) strain at the end of the survival studies (1.6, 3.2 or 10 mg/kg baloxavir acid treatment group). Neutralization antibodies in mice sera were titrated using the hemagglutination inhibition (HI) test [23]. Briefly, the mouse serum was treated with receptor-destroying enzyme (RDE) II reagent (Denka Seiken, Chuo-ku, Tokyo, Japan) according to the manufacturer’s protocol and then was incubated at 56 °C for 30 min to inactivate RDE and its complements. A/PR/8/34 (H1N1) was diluted in PBS to a final HA titre of 8 HA units per 50 µL. Serially-diluted sera were incubated with diluted virus for 1 hr at 4 °C. Next, 0.5% suspension of chicken red blood cells (Japan Bioserum, Fukuyama City, Hiroshima, Japan) was added and incubation was continued for 30 min at room temperature. HI titres were determined as the reciprocal of the highest dilution factor of sera to inhibit hemagglutination of red blood cells.

## 3. Results

### 3.1. Pharmacokinetic Analysis for Subcutaneous Administration of Baloxavir Acid to Mice

Figure 1 and Appendix A show the plasma concentration of baloxavir acid after a single subcutaneous administration to mice. Mice were subcutaneously administered baloxavir acid suspension in 0.5 *w/v*% MC solution (1.6, 3.2 or 6.4 mg/kg). The plasma concentrations of baloxavir acid were determined by LC/MS/MS. The plasma concentration for 10 mg/kg of baloxavir acid came from another report [20]. The mean plasma concentrations were dose-dependent and gradually declined in a time-dependent manner at all administration doses in the same fashion as reported previously for a ferret model [24]. The elimination half-life from 24 to 120 hr for 10 mg dosing in mice was 49.9 h, which was comparable to that of humans (49 to 91 h). The elimination half-life of 6.4 mg dosing and of 3.2 mg dosing (84.2 h and 74.2 h, respectively) in mice were also equivalent to that in humans [20,21]. On the other hand, the concentration–time curve for 1.6 mg dosing in mice was steeper than for the other doses, and the concentration at 194 h after dosing of baloxavir acid at the dose of 1.6 mg/kg was BLQ (<0.1 ng/mL).

### 3.2. Prophylactic Effect of Baloxavir Acid and Oseltamivir Phosphate in Mice Lethally Infected with Influenza A and B Viruses

To evaluate the prophylactic effect of baloxavir acid against lethal infection, mice were administered baloxavir acid at 10 mg/kg once (single dose) or oseltamivir phosphate at 5 mg/kg once daily for 10 days (multiple doses), followed by infection with A/PR/8/34 (H1N1) or B/Hong Kong/5/72 strain at 24 h after the initial dosing. Baloxavir acid treatment completely eliminated mortality due to infection with both A/PR/8/34 (H1N1) and B/Hong Kong/5/72 strains. On the other hand, in the groups administered oseltamivir phosphate, 100% and 70% of the mice survived when infected with A/PR/8/34 (H1N1) and B/Hong Kong/5/72 strain, respectively (Figure 2a,b). Baloxavir acid and oseltamivir phosphate treatment also suppressed body weight loss induced by infection with both virus strains (Figure 2c,d).

To evaluate the antiviral effect of prophylactic treatment with baloxavir acid and oseltamivir phosphate on viral propagation, virus titres in the lungs were examined. When the mice were infected with A/PR/8/34 (H1N1) strain, virus titres in the group treated with baloxavir acid were significantly low at 1, 2 and 4 days after infection compared to the vehicle-treated group (Figure 2e). Virus titres in the group treated with oseltamivir phosphate were also significantly low at 1, 2 and 4 days after infection compared to the vehicle-treated group. Notably, the virus titres in all mice treated with baloxavir acid were at the lower limit of quantification (1.5 log_10_ TCID_50_/mL) during the study period (Day 1 to 10). When infected with B/Hong Kong/5/72 strain, virus titres in the group treated with baloxavir acid were also significantly low at 1, 2 and 4 days after infection compared to the vehicle-treated group (Figure 2f). Virus titres in the group treated with oseltamivir phosphate were comparable to those in the vehicle-treated group at 1 day after infection, but significantly low at 2 and 4 days after infection compared to the vehicle-treated group. As a result, at all evaluation time points, the virus titres in the group treated with baloxavir acid were significantly lower than those in the group treated with oseltamivir phosphate in both viral infection models.

### 3.3. Determination of the Effective Plasma Concentration of Baloxavir Acid in the Prophylactic Treatment Mouse Model

We next used this mouse model to investigate the minimal plasma concentration required to improve survival by prophylactic treatment with baloxavir acid. Several patterns of plasma concentrations at the time of infection were established by administering baloxavir acid at several doses and changing the timing of infection after administration. When 3.2 and 6.4 mg/kg of baloxavir acid were administered at 48 h before infection, its plasma concentrations at the time of infection were 5.45 and 11.2 ng/mL, respectively. When 1.6, 3.2 and 6.4 mg/kg of baloxavir acid were administered at 72 h before infection, its plasma concentrations at the time of infection were 2.25, 3.58 and 8.70 ng/mL, respectively. When 1.6 mg/kg of baloxavir acid was administered at 96 h before infection, its plasma concentration at the time of infection was 0.88 ng/mL (Table 1 and Appendix A).

In the A/PR/34/8-infected mouse model, when baloxavir acid was administered 96 h before infection, the group administered 1.6 mg/kg of baloxavir acid (plasma concentration at the time of infection: 0.88 ng/mL) showed complete elimination of mortality, and a significantly greater prolongation of the survival time compared to the vehicle-treated group. In groups with higher plasma concentrations (2.25~26.1 ng/mL), all mice survived. In the influenza A virus infection model, significant efficacy was observed at the lowest dose (1.6 mg/kg), so efficacy evaluations were not conducted at doses of 3.2 and 6.4 mg/kg. (Table 1, Figure 3a,b).

On the other hand, in the B/Hong Kong/5/72-infected mouse model, when baloxavir acid was administered at 48 or 72 h before infection, all mice survived in the group given 6.4 mg/kg of baloxavir acid. At the dose of 3.2 mg/kg, baloxavir acid also significantly improved survival when administered at 48, 72 or 96 h before infection, whereas the survival rate in the group given 1.6 mg/kg of baloxavir acid was 50% at most (Table 1, Figure 4a–c).

We also examined the inhibitory effect of prophylactic treatment with baloxavir acid on virus replication in the lungs. Virus titres in the groups treated with baloxavir acid were significantly lower than those in the vehicle-treated group by the overall time analysis of mice infected with both A/PR/8/34 (H1N1) and B/Hong Kong/5/72 strains (Appendix A). In baloxavir acid-treated groups, the increase in virus titres was suppressed, and the titres reached around the lower limit of quantification (1.50 log_10_ TCID_50_/mL) by day 10 after infection (Figure 3c and Figure 4d). 

### 3.4. Induction of the Antibody Response

We investigated whether prophylactic treatment with baloxavir acid affects the antibody response induced by the influenza virus infections. In the A/PR/8/34 (H1N1) infection model in which a relatively large viral reduction effect was observed by treatment with baloxavir acid, we considered that its administration at a high concentration might attenuate antibody production. Therefore, we assessed the HI antibody titres in sera of mice infected with A/PR/8/34 (H1N1) strain at 21 to 30 days after infection. The HI antibody titres were 160 to 640 HI in groups treated with baloxavir acid at the dose of 1.6 or 3.2 mg/kg (Table 2). In contrast, few antibody responses were observed in mice treated with baloxavir acid at the dose of 10 mg/kg, with the virus titres being below the limit of quantification.

## 4. Discussion

This is the first study to evaluate the efficacy of prophylactic treatment with baloxavir acid in comparison with oseltamivir phosphate in mice infected with influenza A and B viruses in a dosing regimen that mimics human plasma concentrations. In this study, oseltamivir phosphate was also evaluated as a reference drug, as it is an orally available NAI with relatively abundant clinical evidence regarding prophylactic use [6,7]. The results demonstrated that prophylactic treatment with baloxavir acid significantly reduced mortality of mice infected with influenza A and B viruses. In addition, the prophylactic effect of baloxavir acid on mortality and viral propagation in mice was equal to or greater than that of oseltamivir phosphate. We also investigated the plasma concentrations required for prophylactic efficacy of baloxavir acid in the mouse model. The previously reported BLOCKSTONE trial had few index patients with influenza B and did not include an NAI treatment group. Therefore, our current study offers evidence to complement findings regarding the prophylactic potential of baloxavir acid against influenza.

In humans, baloxavir marboxil (prodrug) is orally prescribed for the treatment of and prophylaxis against influenza. However, as previously reported, the half-life of baloxavir acid (active form) in plasma after oral administration of the prodrug in mice (2.2 to 3.1 h) is much shorter than that of humans (49 to 91 h). In order to increase the accuracy of predicting the prophylactic potential of baloxavir, we investigated the subcutaneous administration of baloxavir acid suspension [20]. In this study, subcutaneous administration of the baloxavir acid suspensions to mice at doses of 3.2, 6.4 and 10 mg/kg achieved plasma half-lives of 74.2, 84.2 and 49.9 h, respectively, and thus these dosing regimens could mimic the human pharmacokinetics of baloxavir acid.

According to previous post-exposure prophylaxis studies in humans, including the BLOCKSTONE trial, most of the household contacts in the placebo group developed influenza symptoms within 5 days [19,25,26,27]. Accordingly, it seems that viral transmission occurred from patients to household contacts within 3 days (72 h) after the contacts had received baloxavir marboxil, with consideration of the incubation period. With the 10 mg/kg subcutaneous dose that we administered to mice in this study, the plasma concentration at 24 h after dosing was 26.1 ng/mL, which was equivalent to that in humans at 1 day (non-Asian, 30.8 ng/mL) to 3 days (Asian, 26.6 ng/mL) after prophylactic treatment. In a mouse model mimicking such a case, baloxavir acid treatment showed significant suppression of viral replication, mortality and body weight loss induced by virus infection. In particular, for A/PR/8/34 infection, the virus titres in the lungs were below the lower limit of quantitation throughout the study period, and no increase in antibody titre was observed. Even with the B/Hong Kong/5/72 infection model, the virus titres in the baloxavir acid treatment group were lower than those in the vehicle and oseltamivir phosphate treatment groups during the study period. They remained at the lower limit of quantification from day 8 onwards. As a result, a significant survival effect was observed by baloxavir acid treatment in mice, suggesting that baloxavir acid can exert a preventive effect against influenza B virus infection as well as influenza A virus infection.

In the current study, several patterns for the plasma concentration at the time of infection were set by adjusting the administration dose and timing of virus infection in order to determine the effective plasma concentration required for influenza A and B virus infection in this mouse model. Our findings indicate that the effective plasma concentration is over 0.88 ng/mL at the time of infection for A/PR/8/34 infection and over 3.58 ng/mL for B/Hong Kong/5/72 infection in this mouse model, which correspond to plasma concentrations on day 19 and 13 after administration of baloxavir marboxil in humans (Asian), respectively. Since there are racial differences in human plasma concentrations based on PK profiles between Asian and non-Asian populations, the effective plasma concentrations for influenza A and B virus infection are equivalent to day 13 and day 8 in humans (non-Asian), respectively (Appendix A).

The effective plasma concentration for influenza B virus is higher than for influenza A virus in the mouse model, which can be explained by the difference in the in vitro activity against each virus type. In our previous study, baloxavir acid showed inhibitory activity against influenza A and B viruses with mean EC_90_ values of 0.46–0.98 nM and 2.2–6.5 nM, respectively. Actually, baloxavir acid inhibited the viral yield of A/PR/8/34 (H1N1) and B/Hong Kong/5/72 strain with EC_90_ values of 0.79 and 2.2 nmol/L, respectively [12]. It was also reported that the mean EC_50_ values of baloxavir acid against seasonal influenza A viruses were five to ten times lower than that of influenza B viruses [13,14,15]. Baloxavir marboxil has been approved and is used for the prevention of influenza A and B virus infections. The prophylactic effect in this mouse model provides supporting evidence that it offers protection against not only influenza A virus but also influenza B virus infection.

Prophylactic treatment with baloxavir acid significantly reduced the lung viral titres in infected mice. These results suggest that prophylactic treatment of baloxavir marboxil can reduce the risk of transmission to further contacts from the person receiving this treatment. Indeed, the previous ferret experiment demonstrated that treatment with baloxavir acid reduced virus transmission [24]. In addition, during the observation period, no viral rebound was observed in this study. From these results, virus replication was almost completely suppressed by the 10 mg/kg treatment in the influenza A virus infection model, and even at lower doses, baloxavir acid may have appropriately inhibited viral replication as well as effectively induced neutralizing antibodies.

In this study, PA/I38T (1 of 141 samples) and PA/A36T (3 of 141 samples) substitutions were found in A/PR/8/34 (H1N1) strain at 2 or 4 days after administration of baloxavir acid (Appendix A). The PA/I38T/F/M/S/N, PA/A37T, PA/E199G or PA/E23G/K/R were detected as treatment-emergent influenza A virus variants with reduced susceptibility to baloxavir acid in clinical use [28,29,30,31,32] and PA/I38T/M or PA/E23K in the BLOCKSTONE trial [19]. The fold change of antiviral susceptibility for PA/A36T was 1.53-fold to the parent virus, suggesting that this substitution has little effect on the susceptibility to baloxavir acid (Appendix A), whereas PA/I38T confers the reduction in baloxavir acid susceptibility. None of the mice, in which these mutant viruses were detected, showed a significant impact on the prophylactic effect of baloxavir acid on virus propagation, body weight or survival. Although the detection frequency of mutant viruses with reduced susceptibility in humans remains low, it is important to continue to monitor the detection status and the susceptibility to baloxavir acid in clinical use.

As we have previously reported, mouse PKPD analysis has confirmed that the plasma concentration (C_tau_ or C_24_) is a predictive parameter for drug efficacy [22]. Based on this PKPD data, clinical trials for therapeutic treatment have been conducted and antiviral effects have been proven [18]. Regarding prophylactic treatment, it is also considered to be important to maintain the plasma concentration in order to exert prophylactic efficacy. As a result, we conducted a clinical trial based on the preclinical data obtained in this study, and this drug was successfully approved for post-exposure prophylaxis against influenza A and B virus infections [19]. Therefore, we believe that the analysis results in this model using an administration method that shows a PK profile mimicking plasma exposure in humans can provide data that support the effectiveness of prophylactic treatment in the clinical setting.

In summary, prophylactic treatment with baloxavir acid in dosing regimens mimicking human PK significantly reduced lung virus titres and eliminated mortality in mice infected with influenza A and B virus, suggesting that prophylactic treatment with baloxavir marboxil may be a useful option for preventing influenza virus infection, regardless of the virus type.

## Figures and Tables

**Figure 1 viruses-15-02264-f001:**
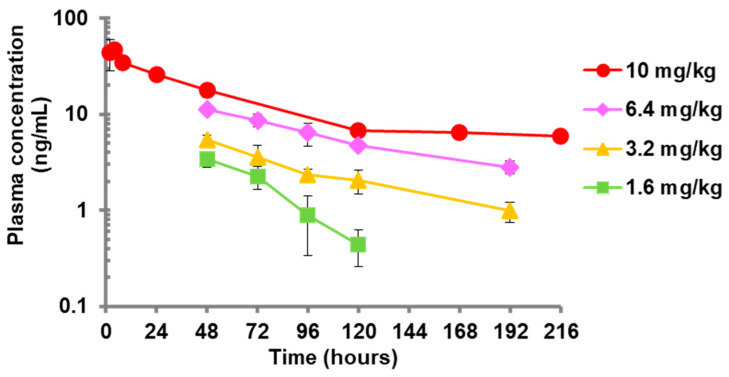
Plasma concentration–time profiles after single subcutaneous administration of baloxavir acid suspension to mice. Four mice were administered baloxavir acid (BXA) subcutaneously. Blood samples were collected at each time point, followed by determination of the concentrations of BXA in plasma. Each plot represents the mean ± SD of 4 mice. The lower limit of quantification (BLQ) is <0.1 ng/mL. The mean values for BLQ were not plotted. The plasma concentrations after treatment with 10 mg of BXA were from another study [20].

**Figure 2 viruses-15-02264-f002:**
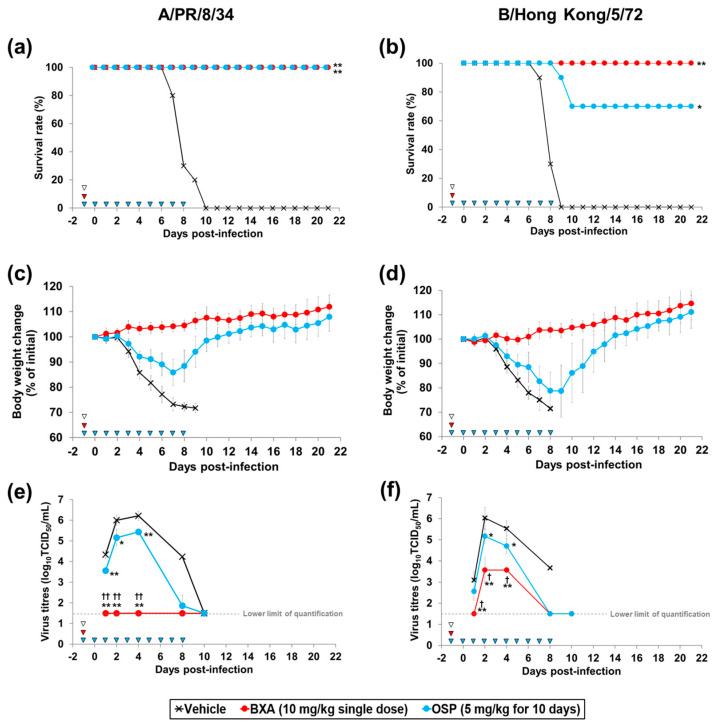
Effect of prophylactic treatment with baloxavir acid and oseltamivir phosphate on mortality, body weight loss and lung virus titre in mice infected with influenza A or B virus. Ten mice were subcutaneously administered baloxavir acid (BXA) or vehicle 24 h before infection. As a reference, from 24 h before infection, mice were orally administered oseltamivir phosphate (OSP) once a day for 10 days. The mice were then intranasally inoculated with A/PR/8/34 (H1N1) or B/Hong Hong/5/72 strain at 24 h after the initial dosing. Survival (**a**,**b**) and body weight (**c**,**d**) were monitored daily for 21 days after the virus infection. Lung virus titres were measured at 1, 2, 4, 8 and 10 days after infection (**e**,**f**). Triangles indicate the time points of administration of each compound. Survival times among the three groups were analysed by log-rank test. For the pairwise comparison of the virus titres in lungs between two groups at each time point, the one-way analysis of variance model was applied by time point. * and ** denote *p* < 0.05 and *p* < 0.0001 vs. the vehicle group. ^†^ and ^††^ denote *p* < 0.05 and *p* < 0.0001 vs. OSP group.

**Figure 3 viruses-15-02264-f003:**
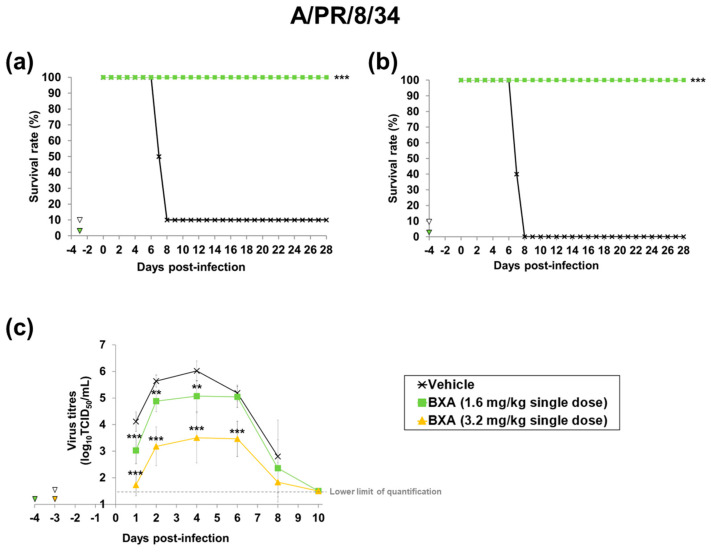
Effect of baloxavir acid at each administration timing and dosage on mortality and lung virus titre in mice infected with influenza A virus. Ten mice were subcutaneously administered baloxavir acid (BXA) or vehicle at 72 or 96 h before infection. The mice were then intranasally inoculated with A/PR/8/34 (H1N1) strain at 72 or 96 h after dosing. Survival and body weight were monitored daily for 28 days after the virus infection (**a**,**b**). Viral titres in supernatants of lung homogenates were determined (*n* = 10 per group) on days 1, 2, 4, 6, 8 and 10 after infection by standard TCID_50_ assay (**c**). Triangles indicate the time points of administration of each compound. Survival times among the three groups were analysed by log-rank test. For the pairwise comparison of the virus titres in lungs between two groups each at each time point, the one-way analysis of variance model was applied by time point. ** and *** denote *p* < 0.01 and *p* < 0.0001 vs. the vehicle group, respectively.

**Figure 4 viruses-15-02264-f004:**
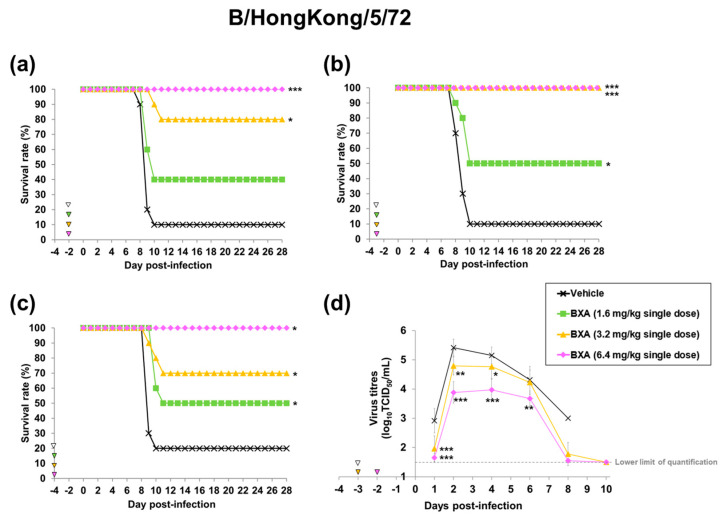
Effect of baloxavir acid at each administration timing and dosage on mortality and lung virus titre in mice infected with influenza B virus. Ten mice were subcutaneously administered baloxavir acid (BXA) or vehicle 48, 72 or 96 h before infection. The mice were then intranasally inoculated with B/Hong Hong/5/72 strain at 48, 72 or 96 h after dosing. Survival and body weight were examined daily for 28 days after the virus infection (**a**–**c**). Viral titres in supernatants of lung homogenates were determined (*n* = 10 per group) on days 1, 2, 4, 6, 8 and 10 after infection by standard TCID_50_ assay (**d**). Triangles indicate the time points of administration of each compound. Survival times among the three groups were analysed by log-rank test. For pairwise comparison of the virus titres in lungs between two groups each at each time point, the one-way analysis of variance model was applied by time point. *, ** and *** denote *p* < 0.05, *p* < 0.01 and *p* < 0.0001 vs. the vehicle group, respectively.

**Table 1 viruses-15-02264-t001:** Plasma concentration of baloxavir acid at the time of infection, and survival rate of mice in each treatment group.

Dose (mg/kg)	Dose Timing (Hours before Infection)	Plasma Concentration at the Time of Infection (ng/mL)	Survival Rate (%)	
			A/PR/8/34	B/Hong Kong/5/72
10	24	26.1	100 **	100 **
6.4	48	11.2	N.T.	100 **
	72	8.70	N.T.	100 **
3.2	48	5.45	N.T.	80 *
	72	3.58	N.T.	100 **
1.6	48	3.39	N.T.	50
	72	2.25	100 **	50 *
	96	0.88	100 **	50 *

Mice were administered baloxavir acid (BXA) suspension, followed by infection with A/PR/8/34 (H1N1) or B/Hong Hong/5/72 strain. BXA concentrations in mice sera are expressed as the mean value of four mice. Survival times among the three groups were analysed by log-rank test. * and ** denote *p* < 0.05 and *p* < 0.0001 vs. vehicle group. N.T., not tested.

**Table 2 viruses-15-02264-t002:** Antibody response of mice infected with A/PR/8/34 strain after prophylactic treatment with baloxavir acid.

Dose (mg/kg)	Dose Timing (Hours before Infection)	Days Post-Infection	HI Titer ± SD (No. of Mice)	
			Calculated	Not Calculated
10	24	21	20 (1/10)	<10 (9/10)
3.2	72	30	320 ± 277 (3/3)	<10 (0/3)
1.6	96	28	368 ± 152 (10/10)	<10 (0/10)

The blood samples were collected from the mice infected with A/PR/8/34 at the end of the survival study. HI test was performed by the standard method as described in Methods section. HI titres were not detected (<10) in negative control (uninfected mice).

## Data Availability

We will share the data upon reviewing the request. Requests of data sharing from researchers will be reviewed with regard to the validity and feasibility of the research intended.

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
