# Peer review of "Prophylactic Treatment with Baloxavir Protects Mice from Lethal Infection with Influenza A and B Viruses"

_viruses, 2023, doi:10.3390/v15112264_

Round 1
Reviewer 1 Report
Comments and Suggestions for Authors
See attached file.

Comments on the Quality of English LanguageEnglish is o.k. Lines 41-43 needs to be reworded: "As neuraminidase..., prophylactic use." This is not a complete sentence.
Reviewer 2 Report
Comments and Suggestions for Authors
In this manuscript, the authors conducted an assessment of the prophylactic efficacy of Baloxavir in inhibiting influenza A and B virus infections in a mouse model. Furthermore, they established a correlation between plasma concentration at the time of infection and the required concentration for virus inhibition. The story is straightforward and well written and important for the drug treatment of the influenza virus. However, the supplementary materials are missing. It would be great if the author could also provide the supplementary documents for further consideration.
Round 2
Reviewer 1 Report
Comments and Suggestions for Authors
On my comment #3 the authors could supply or refer to supplemental data, which contains the independent data set to which the authors refer in their amended text.
On comment #4 the authors could include the text they wrote in their response: "...the difference in administration route is not considered to be a major problem." This would clarify for readers that the authors acknowledge the difference in the route of administration but note that they believe this will not affect their study.
Finally, in comment #5 can the new text that was inserted be substantiated with supplemental data - since the preliminary experiments were already performed?
Comments on the Quality of English Language
None.